# Apoptosis as a Barrier against CIN and Aneuploidy

**DOI:** 10.3390/cancers15010030

**Published:** 2022-12-21

**Authors:** Johannes G. Weiss, Filip Gallob, Patricia Rieder, Andreas Villunger

**Affiliations:** 1Institute for Developmental Immunology, Biocenter, Medical University of Innsbruck, 6020 Innsbruck, Austria; 2Department of Paediatrics I, Medical University of Innsbruck, 6020 Innsbruck, Austria; 3CeMM Research Center for Molecular Medicine of the Austrian Academy of Sciences, 1090 Vienna, Austria; 4Ludwig Boltzmann Institute for Rare and Undiagnosed Diseases, 1090 Vienna, Austria

**Keywords:** chromosomal instability (CIN), aneuploidy, p53, BCL2 family, apoptosis, spindle assembly checkpoint (SAC), cancer

## Abstract

**Simple Summary:**

Errors in the distribution of genetic information, contained in structures referred to as chromosomes, which are made up by DNA and proteins, during a cell division (mitosis) can cause developmental defects and contribute to malignant disease. To prevent the potentially detrimental consequences of unfaithful separation of chromosomes, or parts thereof, a process referred to as chromosomal instability (CIN), cells have developed different solutions. One such solution is the removal of chromosomally instable cells by suicide. This is achieved by the activation of a cell-intrinsic program, termed *apoptosis*. Lost cells are usually replenished by those that achieve the precise distribution of genetic information between daughter cells, in order to maintain a healthy balance in tissues (homeostasis). Mis-segregation of chromosomes reduces cellular fitness, e.g., due to accumulation of DNA damage during the process or imbalances in cellular protein levels arising from inheriting an uneven number of chromosomes (aneuploidy). All these events can trigger activation of the apoptosis machinery. Losing the ability to commit cellular suicide under such conditions can contribute to the rise of cancer, or foster drug-resistance in established tumours during treatment. Here, we discuss the mechanisms involved that promote cellular suicide in cells that show CIN or aneuploidy and how such cells may eventually manage to escape cell death.

**Abstract:**

Aneuploidy is the gain or loss of entire chromosomes, chromosome arms or fragments. Over 100 years ago, aneuploidy was described to be a feature of cancer and is now known to be present in 68–90% of malignancies. Aneuploidy promotes cancer growth, reduces therapy response and frequently worsens prognosis. Chromosomal instability (CIN) is recognized as the main cause of aneuploidy. CIN itself is a dynamic but stochastic process consisting of different DNA content-altering events. These can include impaired replication fidelity and insufficient clearance of DNA damage as well as chromosomal mis-segregation, micronuclei formation, chromothripsis or cytokinesis failure. All these events can disembogue in segmental, structural and numerical chromosome alterations. While low levels of CIN can foster malignant disease, high levels frequently trigger cell death, which supports the “aneuploidy paradox” that refers to the intrinsically negative impact of a highly aberrant karyotype on cellular fitness. Here, we review how the cellular response to CIN and aneuploidy can drive the clearance of karyotypically unstable cells through the induction of apoptosis. Furthermore, we discuss the different modes of p53 activation triggered in response to mitotic perturbations that can potentially trigger CIN and/or aneuploidy.

## 1. Introduction

The faithful and uniform distribution of genomic information into newly emerging daughter cells during mitosis poses several challenges. Multiple complex processes are involved in fulfilling this task. Starting from correct and complete DNA duplication in S-phase, timed entry from G2 into M-phase, chromosome condensation, formation of a mitotic spindle and the attachment of chromosomes, their separation and cell division (cytokinesis), all these processes bear the risk for errors that endanger genome integrity. Hence, cell cycle progression involves passage through multiple checkpoints that prevent chromosome segregation errors and structural CIN. When these checkpoints cannot be satisfied, different fail-safe mechanisms kick in, which prevent the loss of genome integrity at various levels. Here, we aim to review the established links between the cell cycle and apoptotic cell death machinery in response to delayed mitosis, chromosome alterations or defective cytokinesis, acting as a barrier to malignant transformation and therapy-resistance. We emphasize here, that the induction of apoptosis is not the only event that can limit the spread of CIN and aneuploidy in a vulnerable cell population, as the stochastic nature of the process may directly interfere with oncogenic signalling. 

## 2. Mitotic Surveillance by the Spindle Assembly Checkpoint

The major cellular signalling complex guarding against numerical CIN during cell division is the mitotic spindle assembly checkpoint (SAC). The SAC prevents cell cycle progression into anaphase by inhibiting the anaphase-promoting complex/cyclosome (APC/C) E3 ubiquitin ligase complex from engaging its critical coactivator, CDC20. Induction of metaphase to anaphase transition is mediated by APC-dependent ubiquitination and degradation of securin and Cyclin B1, leading to a loss of cyclin dependent kinase (CDK1) activity and separase activation, allowing separation of sister chromatids and mitotic exit. The SAC involves several proteins, first and foremost those that comprise the mitotic checkpoint complex (MCC). The MCC includes MAD2 (mitotic arrest deficient 2) and the mitotic checkpoint proteins BUB3 and BUBR1 that sequester CDC20 away from the APC, thereby preventing the degradation of its mitotic substrates [1]. Importantly, for proper SAC function all proteins have to work in conjunction to induce mitotic arrest when a single kinetochore remains unattached [1]. When the SAC is activated, cells have three possibilities to respond. Cells can either die in mitosis, eventually complete cell division with severe delays (which can lead to chromosomal mis-segregation and CIN), or undergo mitotic slippage into the next interphase without completing cytokinesis. In the latter scenario, cellular or nuclear ploidy as well as the centrosome number increase [2,3,4]. Consequences of polyploidization are discussed towards the end of this review. For a more detailed description of the spindle assembly checkpoint, we refer the reader to excellent reviews on this topic [1,5]. Here, we focus on the connections between the SAC and the mitochondrial pathway of apoptosis.

## 3. Preventing CIN by the Induction of Apoptosis

### 3.1. Mitotic Arrest and Apoptosis

In general, cell death during extended mitotic arrest is executed along the intrinsic *aka* mitochondrial apoptosis pathway (Figure 1). Mitochondrial apoptosis is a highly regulated and largely non-inflammatory cellular suicide program that is executed by the BCL2 protein family, regulating organogenesis, tissue homeostasis and self-tolerance [6,7]. The BCL2 protein family consists of pro-survival proteins (BCLX, BCL2, MCL1, A1/BFL1, BCLB), pro-apoptotic BH3-only proteins (BIM, PUMA, NOXA, BID, BAD, BIK, BMF, HRK) and pro-apoptotic effector proteins (BAX, BAK1, BOK) [7,8]. In steady-state conditions, these proteins balance each other in their activity, but upon stress apoptotic trigger translate into the activation of pro-apoptotic BAX (BCL2-associated X protein) and BAK1 (BCL2 antagonist 1). This can be achieved by transcriptional activation of BH3-only proteins, post-transcriptional mRNA accumulation or different post-translational modifications allowing them to effectively bind and neutralize pro-survival proteins, or activate BAX/BAK directly [9]. Pro-survival proteins themselves can also be repressed, post-translationally modified or degraded, tilting the balance towards apoptosis execution [9]. For more detailed information on this topic we would like to refer the interested reader to the following excellent reviews [10,11]. BAX and BAK dimerization is considered the nucleating event, driving the assembly of mixed higher oligomeric structures and pore formation in the MOM (mitochondrial outer membrane) [12,13]. Upon mitochondrial outer membrane permeabilization (MOMP), cytochrome *c* release enables apoptotic peptidase activating factor 1 (APAF-1) to oligomerize, triggering the recruitment of pro-caspase-9 and its activation within the “apoptosome”. Ultimately, this results in a proteolytic cascade involving additional members of the caspase family (caspase-3, caspase-7), which are cysteine-dependent aspartate-specific proteases, executing this form of regulated cell death [14,15].

As mentioned, mitotic arrest following lack or faulty attachments of sister-chromatids to kinetochores depends on the inhibition of Cyclin B1 degradation [1]. Nevertheless, Cyclin B1 is degraded slowly by non-canonical APC activity in the absence of CDC20 [16]. When Cyclin B1 levels fall below a critical level, cells may eventually slip out of mitotic arrest, despite the SAC not being satisfied [16]. To prevent chromosome mis-segregation and slippage, apoptosis can be induced when caspase activity exceeds a certain threshold within this timeframe. However, the kinetics of caspase activation appear quite heterogeneous when comparing different cell types and cell lines [2]. These observations have been the basis of the competing network hypothesis [17], where the BCL2-regulated caspase activation or APC-mediated Cyclin B degradation pathways are kick-started in parallel, deciding the cell’s fate, which can be either death in mitosis or mitotic slippage [2,17,18]. While APC appears to display a constitutive low level activity in mitosis, the net balance of the BCL2 network needs to be tilted towards MOMP to trigger caspase-activation so that apoptosis can be induced [9]. To this end, CDK1, which forms a complex with Cyclin B1 during mitosis, is able to phosphorylate pro-survival proteins BCL2 (B-cell CLL/Lymphoma 2) and BCLX (B-cell lymphoma extra-large) [19]. Similarly, MCL1 (myeloid cell leukaemia sequence 1) is also targeted by CDK1 and this post translational modification affects its speed of degradation by the proteasome [20,21,22,23]. As such, MCL1 levels influence the time cells arrest in mitosis and can be considered as a “molecular timer” [24]. Multiple studies suggest that MCL1 degradation is the critical driving force in mitotic cell death, aided by NOXA/PMAIP (Phorbol-12-myristate-13-acetate-induced protein 1), a BH3-only protein with a high binding preference for MCL1 [7]. Physical MCL1/NOXA interaction drives degradation of MCL1, which lowers the threshold to mitotic cell death by unshackling pro-apoptotic protein BIM. This can be neutralized by interaction with MCL1, allowing MOMP [25,26]. Multiple E3 ligases, including the APC/C itself, SCF^β-TrCP^, MULE/HUWE1 or SCF^FBW7^, have been implicated in MCL1 turnover, and also during mitotic arrest [20,22,27]. In addition, we noted that degradation of MCL1/NOXA is co-regulated by the mitochondria-residing E3-ligase MARCH5 (membrane-associated ring finger 5), and that lack of MARCH5 sensitizes cancer cells to microtubule targeting agents [28]. Similarly, when interfering with mitosis by depletion of CHAMP1 (chromosome alignment maintaining phosphoprotein), a protein which participates in microtubule attachment to kinetochores [29], cells showed an increase in CIN and cell death. Although CHAMP1-dependent MCL1 stabilization was also not specific for M-phase, increased mitotic cell death was observed upon its depletion [30], similar to findings made upon MARCH5 depletion [28].

While the MCL1/NOXA/BIM axis appears to be most critical to remove mitotically arrested cells as a barrier against CIN in multiple epithelial cancer cell lines [25,26], the BH3-only protein BMF (BCL2 modifying factor), originally described to be a regulator of anoikis [31], was suggested to contribute to death in mitosis of human dermal fibroblasts. When entering mitosis, BMF expression is limited by binding to FOXM1 (forkhead box protein M1) transcription factor [32]. Additionally, FOXM1 upregulates *Cyclin B* transcription to aid mitotic entry [33]. Upon reduction of FOXM1 expression, cells tend to die in mitosis as BMF repression is alleviated [32]. Furthermore, FOXM1 is repressed by p53 upon DNA damage [34,35]. As deficiency of FOXM1 [33] can lead to chromosomal mis-segregation, we can speculate that the described effects may be a consequence of DNA damage, triggering p53 dependent cell death.

The BH3-only protein BID (BH3-interacting domain death agonist) and BAD (BCL2-associated agonist of cell death) [36,37] have also been implicated in mitotic cell death. While the contribution of BAD was only noted upon overexpression, questioning physiological relevance, phosphorylation of BID on serine 66 early upon mitotic entry appears to facilitate its translocation to the mitochondria and priming in colon cancer cell lines, rendering these cells more prone to cell death upon paclitaxel treatment [36]. The kinase responsible for this phosphorylation event remains to be identified.

Activation of BAX and BAK1 are actually thought to be the last BCL2 family members involved in the events promoting apoptosis execution in mitosis. However, recently it has been suggested that they are not simply downstream effectors, waiting for upstream signals to be integrated, but that they play an active role in mitotic cell death by facilitating CDK1-mediated phosphorylation of pro-survival proteins [38]. BAX, and to a lesser degree BAK1, appear to be able to interact with CDK1, targeting it to the outer mitochondrial membrane where it can phosphorylate BCL2 and BCLX to facilitate apoptosis [38]. Phosphorylation of both BCL2 and BCLX is reduced in BAX/BAK double mutant cells. The increased potential of BAX to shuttle CDK1 to the mitochondrial outer membrane (MOM) compared to BAK can be explained by the fact that BAX actually constantly targets mitochondria, but upon insertion into the MOM it is retro-translocated back into the cytosol by interaction with BCLX [39]. It remains to be seen whether phosphorylation of BCLX by CDK1 reduces its capacity to shunt BAX back into the cytosol, leading to its accumulation at the MOM and pore-formation.

It remains difficult to predict if cells slip or undergo apoptosis in situ when they experience mitotic delays. Moreover, although MOMP is frequently seen as a point of no return in apoptosis signalling, not all cells die after caspase activation. In fact, limited MOMP, referred to as minority MOMP, has been shown to lead to genomic instability by the activation of caspase-activated DNase (CAD) [40]. It is worth mentioning that limited caspase activation-induced CAD activity can cause DNA damage specifically at telomeres in mitosis. Protection of telomeres by TRF2 (telomeric repeat binding factor 2) is lost during this process, leading to a DNA-damage response and p53 activation [41,42,43]. Loss of TRF2 sensitises cells to mitotic cell death [44]. However, some cells may still slip into the next interphase, which may be aided by TRF2 overexpression. The consequences of a subsequent G1 arrest after slippage will be discussed further below.

### 3.2. Mitotic Arrest, Metabolism and Mitophagy

One feature frequently linked to mitochondrial quality control, and in tuning mitochondrial cell death, is mitophagy, the clearance of superfluous or damaged mitochondria by autophagy [45]. During mitosis, mitochondria are divided and distributed equally within the daughter cells, a process regulated by the GTPase DRP1 (dynamin-related protein 1), which is phosphorylated and activated by CDK1 during mitosis [46]. In HeLa cells, silencing of DRP1 by siRNA led to a vast increase in cell death during mitotic arrest, accompanied by an increase in mitophagy [47]. Additionally, in a lung adenocarcinoma xenograft model, tumour proliferation decreased upon inhibition of DRP1 [48]. In both cases, the effects noted did not seem to be linked to the fission-promoting function of DRP1, but may be a consequence of corrupted mitochondrial metabolism, as mitosis is highly energy consuming. When entering mitosis, CDK1 tunes cellular metabolism that becomes highly dependent on mitochondrial respiration. A fraction of Cyclin B/CDK1 can be found in the mitochondrial matrix and phosphorylate components of complex I of the respiratory chain [49]. During extended mitotic arrest, mitochondria are cleared by mitophagy and glycolysis becomes the main energy source for ATP production. This response depends on AMPK (AMP-activated protein kinase) and PFKFB3 [6-phosphofructo-2-kinase/fructose-2,6-biphosphatase 3) ramping up glycolysis for survival [50,51]. Of note, glucose-deprivation may again promote cell death by engaging the BH3-only protein NOXA [52]. As such, it can be hypothesized that loss of mitochondria via mitophagy as well as inhibition of glycolysis should render cells more susceptible to mitotic inhibitors during cancer therapy.

Another metabolism-regulating target of CDK1 and AMPK during mitosis is RAPTOR (regulatory associated protein of mTOR), which is phosphorylated during mitosis, suppressing its activity [53,54]. RAPTOR forms a complex with mTOR called mTORC1 (mammalian TOR complex). The TOR (target of rapamycin) kinase pathway plays an essential role in nutrient response, cell growth and proliferation [55]. Although mitosis is a highly nutrient dependent process, mTORC1 levels are generally low, as mTOR per se does not seem to be necessary for mitotic progression or during prolonged mitotic arrest [56]. mTORC1 is inhibited by RAPTOR phosphorylation on multiple sides [53]. Nevertheless, expression of a RAPTOR phosphorylation mutant in HeLa cells prevented the loss of mTORC1 and cells were less prone to cell death during prolonged mitosis. Cell survival was secured by an increase in anti-apoptotic BCLX levels and a decrease in PDCD4 (programmed cell death protein 4). This suggests that cells able to sustain mTORC1 function in M-phase may be more likely to slip after mitotic arrest, and that mTORC1 downregulation during mitosis may be essential for impeding CIN [56]. Dual mTORC1/2 inhibition is currently under investigation in combination with paclitaxel in high-grade serous ovarian carcinomas which carry MYC amplifications [57].

## 4. Cell Death or Survival in Response to CIN—All about Bribing p53?

p53 is one of the most studied proteins and the plethora of its effector functions apparently needed for active tumour suppression have clouded its precise mode of action. Identification of relevant transcriptional outputs is challenging and the existence of multiple p53 isoforms as well as interaction partners make therapeutic exploitation of our vast knowledge difficult [58,59]. As the modalities and consequences of p53 activation differ substantially between cell types and tissues, many studies limit themselves to using p53 activation as a surrogate readout for tumour suppression, senescence or cell death initiation [59,60,61]. Notably, multiple different pathways leading to p53 that are critical to preventing CIN and aneuploidy have been characterised in recent years (Figure 2).

### 4.1. p53 Activation Due to Chromosomal Mis-Segregation

p53 plays a major role in the DNA damage response and can promote cell death. There is a clear correlation between aneuploidy and *Tp53* mutations across cancers [62,63] and it has been shown that deletion of p53 promotes aneuploidy induction [51,64], whereas aneuploidies do not unequivocally lead to p53 activation [65]. Therefore, p53 seems to play a major role in restraining or eliminating cells with a high rate of CIN. Activation of p53 is likely not due to aneuploidy per se. Instead, events preceding aneuploidy lead to a p53 response and p21-dependent cell cycle arrest in the next G1 phase. DNA damage, as a consequence of chromosomal mis-segregation, DNA-double-stranded breaks or micronuclei formation, and chromothripsis are contributing [66,67,68]. When lagging in mitosis, entire chromosomes, or bigger chromosomal fragments, are initially sealed in separate micronuclei. Micronuclear membranes, however, often lack lamin B or nuclear pores, are prone to rupture, and, upon re-entering S-phase, massive DNA damage is acquired. The latter can be due to insufficiency in DNA building blocks or relevant enzymes. Additionally, those chromosomes can potentially rearrange upon micronuclear breakdown, a phenomenon called chromothripsis [68,69,70]. Both micronuclear envelope breakdown and chromothripsis can induce p53, but also sterile inflammation (see below).

Interestingly, Histone H3.3^Ser31^ phosphorylation on lagging chromosomes during anaphase can also trigger p53-dependent G1 arrest upon chromosomal mis-segregation in the absence of DNA damage [71]. Precisely how aneuploid G1 cells that carry serine phosphorylated H3.3 can activate p53 remains to be established. It will be interesting to test if the somatic mutations in H3.3. found in paediatric glioblastoma patients functionally interfere with this type of aneuploidy checkpoint [72,73]. The consequences of p53 activation beyond p21 induction in such cells also needs to be established, but defective cell death pathway engagement may contribute to the rise of MYCN amplified malignant disease.

### 4.2. P53 Activation in Light of Numerical CIN-Induced Protein Imbalance

Protein aggregation leads to an overflow of proteins at the endoplasmic reticulum (ER) and consequently insufficient protein folding. ER stress initiates a number of signalling events culminating in the unfolded protein response (UPR). The UPR adjusts protein load within the cell by inhibiting ribosomal translation, giving way to more sufficient protein folding [74,75].

In budding yeast, aneuploidy-induced protein aggregation, either due to imbalance of protein complex stoichiometries or poor lysosomal aggregate clearance, leads to proteotoxic stress and proliferation defects. This phenotype is random and not dependent on the accumulation of specific chromosomes or the total number of aneuploid chromosomes [76,77,78]. In mammalian cells, the UPR is strongly interconnected with autophagy. Global transcriptome and proteome analysis in human HCT116 and RPE1 cell lines harbouring extra chromosomes have shown that autophagy is induced in a p62 (sequestosome1; ubiquitin-binding protein p62) specific manner [79]. Additionally, upon numerical CIN induced by SAC inhibition, HeLa cells showed high levels of ER stress and concomitant autophagy induction depending on p62 [80]. P62 acts as a cargo receptor for non-functional proteins that become ubiquitinated and further associates itself with LC3 (microtubule associated light chain 3) to ensure formation of the autophagosome, its conjugation with lysosomes and autophagolysosomal degradation [81,82]. It was also shown that p53 induction accompanies the UPR and p62 enrichment in those cells. The modalities leading to p53 stabilization in this setting were not explored, but may involve PERK mediated alternative start site activation in the *p53* gene locus, or sequestration of MDM2 by ribosomal proteins [83]. As a result, cells died by apoptosis upon protein accumulation in a dose-dependent manner [80], further underlining the interaction between aneuploidy, autophagy and the apoptotic machinery. Induction of apoptosis in the context of ER stress has been linked to the BCL2 family proteins BIM and PUMA [84], but a role for these proteins in eliminating aneuploid cells experiencing ER stress remains to be established.

Interestingly, although drugs causing proteotoxic stress lead to p53 activation and apoptosis in trisomic MEFs (mouse embryonic fibroblasts), the same drugs showed antiproliferative effects on aneuploid cell lines independent of p53 mutation status [85]. Nevertheless, DNA damage induced by doxorubicin increased autophagy and cell death in trisomic ES cells (embryonic stem cells) and cell death was alleviated by the inhibition of autophagy [86], suggesting interconnection of these pathways. Recently, Singla et al. reported that, in mouse embryos, aneuploidy induced by SAC inhibition leads to apoptotic cell death of aneuploid cells in the epiblast during early embryogenesis, and that removal of aneuploid cells triggered compensatory proliferation of diploid cells [87]. Both p53 and LC3 were found to be upregulated in aneuploid embryos, while *Bcl2* mRNA levels were reduced, limiting proliferation of those cells. Inhibition of p53 resulted in decreased activation of autophagy, whereas inhibition of autophagy had no influence on *p53* mRNA levels. This suggests an interconnection between p53 and autophagy rather than two events occurring in parallel, as here p53 activation due to aneuploidy triggers autophagy and this drives elimination of aneuploid cells by apoptosis [87]. While the induction of autophagy in aneuploid cells appears as a more general response, the requirement for p53 in this process differs, as upon chromosomal mis-segregation in RPE1 cells, lysosomal activation was comparable between p53-proficient and deficient cells [88]. Although certain facts point to a possible p53-dependent activation of autophagy in response to CIN and aneuploidy, more work needs to be carried out to demonstrate a direct interaction and to dissect the effectors of cell death involved in the elimination of aneuploid cells that engage autophagy prior to apoptosis.

### 4.3. P53 Activation Due to Delayed Mitotic Progression

Cells experiencing delays passing through mitosis, e.g., caused by delayed satisfaction of the SAC or due to defects in centriole biogenesis, also induce a p53 response when entering the next G1 phase. While in cell line studies p21-induced cell cycle arrest appears the primary response to centriole loss, acentriolar mitoses caused by spindle assembly abnormal 4 (SAS-4) deletion promote apoptosis in the developing mouse embryo [89,90]. Importantly, p53 activation appeared to be independent of DNA damage, suggesting the existence of a “centrosome surveillance” pathway to limit the growth or survival of cells that experience errors in the centrosome duplication cycle [89,90]. Loss of centrioles interferes with mitotic timing [90,91] and an early study also reported the phenomenon that a prolonged prometaphase blocks proliferation, suggesting the existence of a “mitotic stop watch” [92]. Three studies eventually provided a mechanistic explanation for how p53 becomes activated in cells that experience mitotic delays, independent of the type of perturbation [93,94,95]. In one of those, a CRISPR-based genetic screen identified 53BP1 and the deubiquitinase USP28, along with the E3 ligase TRIM37, as key regulators of p53 stabilisation upon centriole depletion and acentriolar proliferative capacity, respectively [94]. Here, 53BP1 appears to act independently of DNA damage and exerts a scaffold function, allowing interaction of p53 with USP28 that antagonizes MDM2-regulated ubiquitination [95]. How 53BP1 is activated in order to exert its function in the mitotic surveillance pathway remains to be fully established but a role for polo-like kinase 1 (PLK1) (https://doi.org/10.1101/2022.11.14.515741, accessed on 1 December 2022) and p38MAPK kinase signalling have been proposed [92,96].

The apoptosis-inducing potential of p53 in this response has been best documented in early embryogenesis, where loss of SAS-4, abrogating centriole biogenesis, causes premature lethality in utero. Co-deletion of USP28, however, allows embryogenesis to proceed significantly further [97]. Similarly, loss of USP28 can rescue neuronal progenitor cells from p53-induced apoptosis in mice that carry mutations found in patients developing microcephaly due to defects in centriole biogenesis that also cause mitotic delays [98]. Upon DNA damage, p53 is able to directly target pro-apoptotic BCL2 proteins, mostly PUMA (p53 upregulated modulator of apoptosis), NOXA and BAX, leading to MOMP, cytochrome c release and the induction of the apoptotic cascade [99]. How p53 actually triggers cell death in the context of the mitotic surveillance pathway has not been addressed, but hematopoietic cells appear to prefer cell death initiation over cell cycle arrest upon PLK4 inhibition (personal observations). As such, targeting USP28 in cancer, with the aim to block its role in DNA repair function, may become a boomerang as it may allow the survival of cells prone to CIN after experiencing delays in mitosis. Moreover, loss of p53 cannot fully restore embryonic development in SAS-4 mutant mice [97], suggesting alternative players to be involved in removing cells facing CIN and at risk for aneuploidy in response to mitotic delays. Finally, while loss of the 53BP1-USP28-p53-p21 axis allowed for the outgrowth of cells facing centrosome loss [95], it did not seem to be involved in limiting the growth of cells that experience cytokinesis failure or centrosome amplification after slippage [94].

### 4.4. p53 Activation by Extra Centrosomes

Accumulating evidence suggests that mitotic slippage (discussed in the context of prolonged SAC activation above), as well as failed cytokinesis, e.g., due to lagging chromosomes or persisting DNA bridges, greatly contribute to the emergence and evolution of cancer cells by promoting polyploidy and a concomitant numerical centrosome amplification. This paves the way for CIN and aneuploidy, which are prominently featured in many cancers. The increase in centrosome number disrupts the organization of the mitotic spindle and bipolar chromosomal segregation. Notably, if mitosis is allowed to proceed in a cell with supernumerary centrosomes, this can result in the formation of a pseudo-bipolar spindle due to the clustering of centrosomes. Mis-attached chromatids lead to chromosome mis-segregation, and this process yields two aneuploidy daughter cells with a variable number of centrosomes. Alternatively, scattered supernumerary centrosomes can lead to a multipolar anaphase. Due to the formation of multiple cleavage furrows, the resulting daughters display a high degree of aneuploidy and are rarely viable. The importance of centrosome number surveillance as an onco-suppressive mechanism is clearly highlighted by the high number of aberrant karyotypes found in tumours [100,101,102].

Previous research has shown that, in the presence of supernumerary centrosomes, cells activate a multi-protein complex called the PIDDosome, which prevents cells from re-entering the cell cycle [103]. Fava et al. demonstrated that the PIDDosome acts as a sensor for the centrosome number, thus controlling ploidy levels in mammalian cells [104]. The PIDDosome comprises the death-domain (DD) containing proteins PIDD1 (P53-Induced Death Domain Protein 1) and RAIDD (RIP-Associated ICH1/CED3-Homologous Protein with Death Domain), as well as Caspase-2, an aspartate-specific endopeptidase [105]. Upon assembly, the PIDDosome facilitates proximity-induced dimerization and autoproteolytic activation of Caspase-2. Several lines of evidence suggest that activated Caspase-2 can contribute to apoptosis and cell cycle arrest in the context of different mitotic perturbations [106,107,108], but its link to centrosome surveillance appears most robust, thus also establishing a link to its reported role as a tumour suppressor wherein it may act by removing aneuploid cells [109,110,111].

Interaction of PIDD1 with extra mature centrosomes, mediated by the distal appendage protein, ANKRD26, promotes PIDDosome assembly in polyploid cells after cytokinesis failure [112]. As a result, Caspase-2 starts cleaving its substrates, most notably MDM2, an E3 ubiquitin ligase that acts as a negative regulator of p53. This leads to p53 stabilization and the transcription of p53 target genes, including *p21*, facilitating cell cycle arrest and preventing the outgrowth of polyploid cells and the onset of CIN [104,113]. As with loss of *p21*, cells deficient of any of the PIDDosome components display impaired cell cycle arrest after cytokinesis failure, establishing a clear link between the acquisition of multiple centrosomes and a p53-dependent cell cycle arrest in cancer cells forced to fail cytokinesis, and in primary hepatocytes undergoing scheduled cytokinesis failure [104,114].

Beyond its role to facilitate cell cycle arrest in a PIDDosome dependent manner, Caspase-2 has frequently been reported to engage MOMP and thus induce apoptosis, e.g., in response to DNA damage or spindle poisons [106,107,108]. Whether Caspase-2 can clear cells with extra centrosomes has not been addressed. How Caspase-2 promotes MOMP has also been a matter of debate [108,115]. Notably, it has previously been reported that Caspase-2 is able to cleave the BCL2 family protein BID to its truncated, pro-apoptotic form tBID [107,116]. However, the substrate specificity of Caspase-2 for BID is low and primary cells from Caspase-2 deficient mice do not display severe apoptotic defects after mitotic perturbations [117]. While evidence accumulates that Caspase-2 might act as a barrier against numerical CIN in cells with extra centrosomes, the mechanistic details remain to be elucidated.

## 5. Preventing the Spread of CIN by Sterile Inflammation

Chromosome mis-segregation associates with micronuclei formation. Due to reduced stability of Lamin B in the nuclear envelope wrapping chromosomes that fail to integrate in the main nucleus, dsDNA becomes detectable in the cytoplasm [118,119]. cGAS (cyclic GMP-AMP synthase) binds cytosolic DNA, leading to the generation of a second messenger cyclic dinucleotide, cGAMP (cyclic guanosine monophosphate—adenosine monophosphate), which binds to STING (stimulator of interferon genes). Although it is an evolutionary conserved pathway against viral or bacterial invasion, DNA-binding of cGAS is not sequence specific and hence not limited to dsDNA from pathogens [120,121]. For cGAS-STING activation, size matters, as more cytosolic DNA leads to a stronger inflammatory response [122]. Additionally, STING can trigger apoptosis in certain cell types directly [123], as well as engaging senescence [124], while initiation of apoptosis itself may again amplify immunologic recognition in a cGAS/STING-dependent manner via mitochondrial DNA (mtDNA) release into the cytosol [12]. Hence, CIN is a very potent trigger of STING-dependent inflammation, and RPE1 cells did show a differing immune response gene pattern after chromosomal mis-segregation, when compared to cells arrested due to DNA damage [125]. Of note, aneuploid cells arrested in G1-phase show an upregulation of cell surface ligands, increasing immunological visibility, rendering these cells more susceptible to NK-cell (natural killer cell) attack. In co-culture systems, aneuploid cells were rapidly eliminated. Contrary to that, euploid cells did not show increased cell death [125].

Spinning the ball back to an earlier topic, there also seems to be a connection between cGAS-STING activation and autophagy. It would be easy to conclude that both pathways are independently activated in response to aneuploidy, but autophagy also appears to be required to dispose of cGAS-activating dsDNA to switch off the signal [126]. Additionally, Krivega et al. showed that, in constitutively trisomic cell lines, upon deletion of cGAS and STING, the expression of TFEB (transcription factor EB), which is the master regulator of lysosomal biogenesis, and its target genes (LC3, p62, LAMP2) were reduced, when compared with diploid cells. The same results could be shown in trisomic human embryonic fibroblasts, suggesting that cGAS-STING is not only responsible for the induction of inflammation after CIN, but also transcriptionally upregulates autophagy in response to aneuploidy, as shown in cells from humans with Down syndrome [127].

Since this inflammatory response to aneuploidy and CIN seems to be genoprotective, e.g., by fostering NK cell recognition, it is puzzling that cGAS and STING, while often lowly expressed in cancer cell lines, are rarely mutated (<1%) in cancer [128]. This is consistent with recently published data, documenting that patient-derived triple negative breast cancers exhibiting high CIN showed higher cGAS activity. Additionally, they were more likely to form metastasis than those with low CIN. Interestingly, STING-dependent NF-κB signaling seemed to positively affect tumour metastasis. Depletion of STING or NF-κB inhibition limited metastatic capacity, while in cancers with low CIN the addition of cGAMP improved metastasis capacity [129]. Upon deletion of cGAS using CRISPR-Cas9 or upon chemical inhibition of cGAS, several human and mouse breast cancer cell lines (BT594, 4T1) showed higher rates of apoptosis upon chromosomal mis-segregation. The same was true for STING inhibition, in line with NF-κB signalling inducing the expression of several pro-survival proteins, e.g., within the BCL2 family. In vivo tumour growth of the chromosomally instable murine 4T1 breast cancer cells was clearly reduced when lacking cGAS or STING [130], indicating that STING activation in the context of CIN is not onco-protective per se. These data urge caution for the use of STING agonists in the context of cancer therapy.

## 6. Conclusions and Outlook

The role of CIN and aneuploidy in cancer formation and evolution is firmly established. The relevance of apoptosis or other cell death modalities as a barrier against CIN or the removal of aneuploid cells, as compared to the induction of cell cycle arrest and senescence, is less clear. Moreover, while multiple pathways have been delineated that lead to p53 activation in response to CIN and aneuploidy, its outcome has mostly been investigated in model cell lines. The ability of p53 to induce cell cycle arrest (targeting p21) or cell death (targeting PUMA, NOXA) for tumour suppression in vivo, however, has been challenged in its importance [59]. This begs the question: are we looking in the right direction? Clearly, it remains to be seen if all routes of p53 activation subsequent to CIN lead to the same transcriptional core signature, canonical apoptosis or inflammatory outcome in different cell types and tissues. Anecdotal evidence of a direct pro-apoptotic role of p53 on mitochondria deserves mentioning in this context [131]. The long-lasting conundrum as to why different cell types elicit different cell fates in response to p53 activation is still largely unanswered. Advances in sequencing technologies, high-resolution and live-cell imaging approaches will shed light about the plasticity of the responses that limit CIN and aneuploidy at the cellular and organism-wide level.

## Figures and Tables

**Figure 1 cancers-15-00030-f001:**
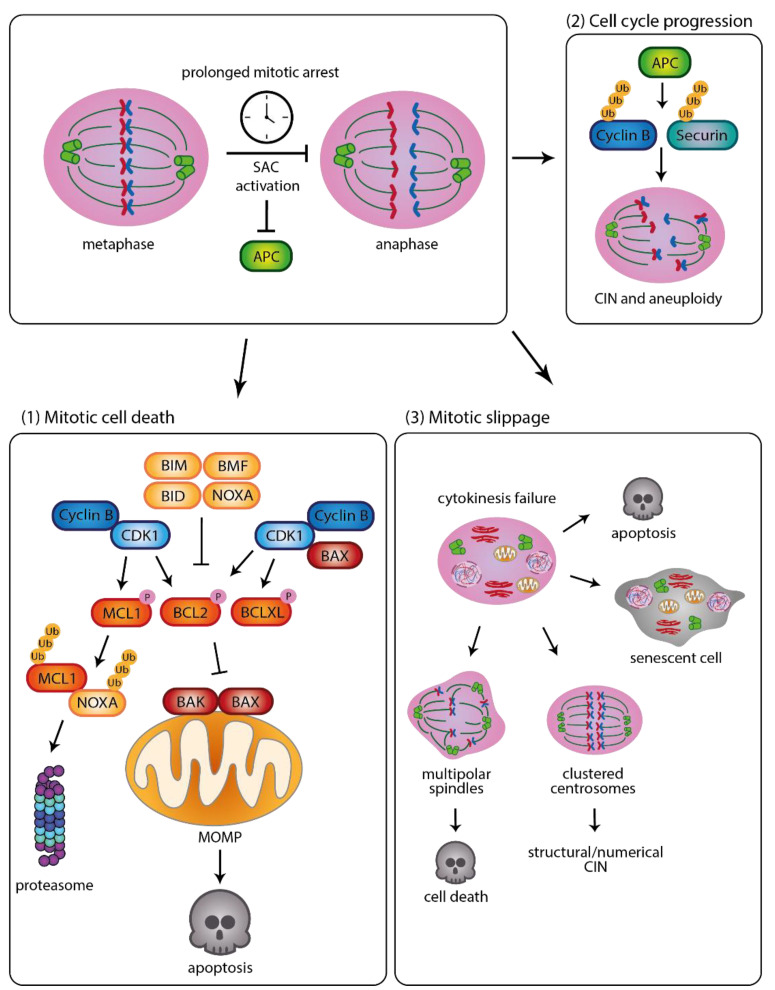
Proper distribution of chromosomes during mitosis is guarded by the spindle assembly checkpoint (SAC) which, upon misguided or missing microtubule attachment, inhibits the anaphase promoting complex/cyclosome (APC/C), the E3 ubiquitin ligase degrading Cyclin B1 and securin. Upon mitotic arrest cells may either (**1**) undergo cell death in mitosis, executed by BCL2 protein family interactions, (**2**) complete cell division with delays, resulting in chromosomal mis-segregation and improper DNA content distribution in daughter cells or (**3**) fail mitosis and exit into the next interphase in a process referred to as mitotic slippage. The resulting whole genome duplication (WGD) of cellular DNA-content and centrosome imbalance can lead to different outcomes. These include apoptosis, senescence or cell cycle re-entry that can lead to structural and/or numerical CIN, when extra centrosomes cluster, or multi-polar spindle formation leading to severe aneuploidies and subsequent cell death that may not need to be strictly apoptotic.

**Figure 2 cancers-15-00030-f002:**
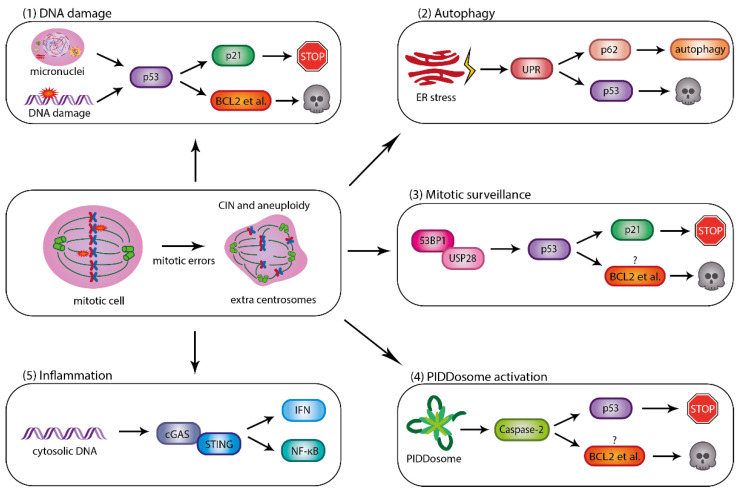
Consequences of chromosomal mis-segregation or mitotic slippage are diverse. (**1**) p53 activation due to DNA damage, e.g., emanating from fragile micronuclei can trigger p21 dependent cell cycle arrest or BCL2 protein family dependent apoptosis. (**2**) ER stress and subsequent unfolded protein response (UPR) as a result of aneuploidy-induced protein imbalances or aggregation can trigger autophagy, involving p62, and/or result in p53 activation leading to apoptotic cell death. (**3**) Delays in mitotic progression (>90 min) can trigger a DNA damage-independent p53 response, involving p53 binding protein 1 (53BP1) and ubiquitin specific protease 28 (USP28), limiting MDM2-dependent p53 degradation. Cells may either arrest or die, potentially in a BCL2-regulated manner. (**4**) Similarly, independent of DNA damage, amplification of centrosomes triggers PIDDosome formation, MDM2 cleavage, p53 stabilization and p21 dependent cell cycle arrest. If and how extra centrosomes trigger apoptotic cell death remains uncertain, but it may involve again the BCL2 family. (**5**) Aggregation of cytosolic dsDNA, e.g., as a result of micronuclei formation, leads to cGAS-STING dependent sterile inflammation, resulting in interferon (IFN) production or induction NF-κB signalling, promoting cell recognition and clearance.

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
