# Peer review of "Apoptosis as a Barrier against CIN and Aneuploidy"

_cancers, 2022, doi:10.3390/cancers15010030_

Round 1
Reviewer 1 Report
This manuscript reviews the interrelation between the mechanisms leading to aneuploidy and numerical CIN in cancer and mitotic cell death processes. This is a very interesting topic since these relationships may be exploited therapeutically. However, there are several concerns regarding this manuscript rendering it not acceptable for publication in “Cancers” at this present form.
Major concerns:
Despite the requirement of thorough editing for English language throughout the manuscript, a major issue involves the definition of CIN that according to the text may be erratically perceived to involve only genome dosage imbalances in the form of whole chromosome or chromosome arm gains or losses (aneuploidy). The authors must clarify that CIN is an ongoing stochastic process of structural and numerical alterations of chromosomes or sets of chromosomes due to replicative stress and insufficient DNA damage responses that in addition to aneuploid or polyploid cells, eventually may generate segmental aneuploidy and structurally altered chromosomes. A subset of these alterations may provide selective advantages to cancer cells and may be clonally maintained during cancer genome evolution, while others may be eliminated from the cancer cell populations but not necessary through radical processes of cell death (i.e.: because they impair oncogenic processes). Another important issue stems from the misuse of the Scientific Terms mitotic slippage and centrosome amplification. Clustered amplified centrosomes due to endoreduplication (a process that is very frequent in cancer genome evolution) do not necessarily lead to tripolar or tetrapolar anaphases, causing massive chromosomal segregation errors but they can promote the emergence of polyploid cancer clones that sustain continuous growth.
Minor concerns:
The whole abstract needs to be rephrased. Examples of unclear phrases: Lines 16-17: “… Chromosomally instable cells need to be kept in check, or cleared, to prohibit the sampling of aneuploid karyotypes able to drive tumorigenesis…” Lines 18-20: “…Here, we review how CIN can be prevented or limited to spread by the induction of cell death and the relevance of different p53 responses triggered in response to mitotic perturbations to prohibit the formation of cancer driving aneuploidies…”
Lines 26, 27: The sentence “…The faithful duplication of genomic information during each cell division cycle and physical its separation into newly emerging daughter cells during mitosis pose several challenges in order to maintain genome integrity…” is unclear and better be rephrased.
Line 30: The term whole genome duplication (WGD) is used to denote polyploidization processes. The phrase “…upon completion of whole genome duplication…” must be corrected to “…upon completion of whole genome replication…” or “…upon completion of genome replication…”
Lines 38-41: This long sentence is unclear and better be rephrased.
Lines 42-45: The sentence is unclear and better be rephrased.
Lines 47: The phrase “…The major cellular signaling complex guarding against CIN…” better be rewritten as “…The major cellular signaling complex guarding against numerical CIN…”
Figure 1 (3): A multipolar mitosis is indicated as the outcome of polyploidization due to mitotic slippage and endoreduplication, but this rather represents polyploidization due to cytokinesis failure or cell–cell fusion.
Lines 136-137: The sentence “…As FOXM1 is an important mitotic regulator and deficiency can lead to chromosomal mis-segregation, observed effects may be lingering on 137 cell death due to impaired mitosis and subsequent p53 responses…” better rephrased.
Line 155: better change “…cytoplasmatic…” to “cytoplasmic”
Figure 2: Micronuclei better be depicted in the first panel.
Lines 227: The phrase: “…can trigger autophagy regulated by p62 and result in p53 activation leading to cell death…” better be changed to “…can trigger autophagy regulated by p62 or result in p53 activation leading to cell death…” similar with Lines 228-229.
Lines 240-241: The sentence is unclear and better be rephrased.
Lines 274-275: Unclear statement
Lines 341-348: These statements need to be re-examined and rephrased taking into consideration the differential production of clustered or scattered supernumerary centrosomes (see major concerns).
Line 435: avoid the word “superficially”
Lines 435-437: The sentence is unclear and better be rephrased.
Author Response
We would like to thank this referee for the time taken to read our contribution and the valuable suggestions, as well as justified points of criticism.
The authors must clarify that CIN is an ongoing stochastic process of structural and numerical alterations of chromosomes or sets of chromosomes due to replicative stress and insufficient DNA damage responses that in addition to aneuploid or polyploid cells, eventually may generate segmental aneuploidy and structurally altered chromosomes.
This comment is important and we have made an effort to point out the nature of CIN more clearly in the re-written abstract (line 30-36)
A subset of these alterations may provide selective advantages to cancer cells and may be clonally maintained during cancer genome evolution, while others may be eliminated from the cancer cell populations but not necessary through radical processes of cell death (i.e.: because they impair oncogenic processes).
We now highlight this fact at the end of the end of the introduction paragraph
Another important issue stems from the misuse of the Scientific Terms mitotic slippage and centrosome amplification. Clustered amplified centrosomes due to endoreduplication (a process that is very frequent in cancer genome evolution) do not necessarily lead to tripolar or tetrapolar anaphases, causing massive chromosomal segregation errors but they can promote the emergence of polyploid cancer clones that sustain continuous growth.
We hightlight the fact that extra centrosomes do not necessarily trigger multi-polar mitoses due to clustering. lines 394-400.
We have also addressed the minor issues, according to suggestions and adopted Figures 1 and 2.
We hope this referee will now support publication of our review
Reviewer 2 Report
The subject matter and scientific content of the review are excellent and will be of interest to a wide readership. The authors have covered both the existing literature but also highlighted questions yet to be answered in the field. However, the paper need significant editing of the text to make it easier to read, there are numerous extremely long sentences and grammatical errors which need to be addressed prior to publication.
The review would benefit from a more concise introduction to the different models of regulation of the BCL-2 family and how the caspases are activated downstream of these proteins. Additionally, as p53 is central to this review it would also be beneficial to introduce its non-transcriptional role in apoptosis as this would help the reader in the later sections. Furthermore, I was surprised the authors chose not to discuss the literature which has characterised the BAX/BAK ‘pore’ from the groups of Ruth Kluck and Anna Garcia-Saez.
Furthermore, as the review focuses on the p53-p21 axis it would be helpful if the authors acknowledge that this axis is also important in Aneuploidy-induced senescence.
Finally when discussing the cGAS-STING pathway, it would be useful for the authors highlight the links of this pathway to both necroptosis and senescence.
Overall, this has the potential to be an excellent review.
Author Response
We would like to thank this referee for the overall positive evaluation of our contribution and the time taken to read and comment on our work.
We have solicited a native speaker for editing the manuscript. We hope this has increased the quality and flow to make it an easier read.
While we now summarize how the caspase cascade becomes activated downstream of MOMP, we believe that introducing the long standing discussion how BCL2 family members engage MOMP via e.g. MODE 1 vs. 2, direct or indirect activation models, will dilute the focus of this review. However, we now refer to recent reviews on this topics, by the true experts in this field. https://doi.org/10.1038/cdd.2017.179 & PMID: 30655609 to guide the interested reader towards a more comprehensive discussion.
We now cite work from the Garcia-Saez lab, reporting on mixed BAX/BAK oligomers this year in Molecular Cell. We now also add work by Ruth Kluck and colleagues (PMID: 28630157), but do not want to engage into the mechanistics of BAX/BAK pore formation, as we feel this is beyond the scope of this review.
While direct cell death-inducing properties of p53 at mitochondria have been reported in the past, none were discussed in the context of mitotic perturbations, nor am I a firm believer in such transcription-independent apoptotic roles. To balance this bias, we now include a reference in the outlook section that summarizes this early work (PMID: 27308326).
We also cite a recent review by the Schmitt lab on the basis of cellular senescence and its link to p53 (PMID: 30602768) and its multiple tumor suppressive effects (PMID: 35396345) at the beginning of the chapter on page 6.
Finally, we discuss the link between cGAS/STING, cell death and senescence, by referring to two papers from Andrea Ablasser´s group, PMID: 28759028 and PMID: 28874664. The potentially indirect link of STING to necroptosis was seen as to complex to be integrated in a simple and meaningful way.
We hope that these amendments will make our review now acceptable for publication.
Reviewer 3 Report
This is a very nice and comprehensive reviw about the pathways that prevent chromosomal instability and aneuploidy. I have only minor suggestions for changes/clarifications.
Abstract line 16. It should be "unstable" not "instable"
Introduction line 27. It seems there is an extra word that needs to be moved after "its" (physical)
Line 111. "degraded" should read "degradation"
Line 242. Can you add a sentence to explain how micronuclei induce p53. A fully formed micronucleus will not trigger p53 activation. Please accurately describe which step in micronucleus formation likely leads to p53 activation.
Author Response
We are thankful for this overall very positive evaluation of our contribution and the time taken to review it.
We have corrected the minor errors, subjected our manuscript to editing by a native speaker and now describe the process how micronuclei accumulate DNA damage in more depth (Lines 272 - 283). We hope that these changes will be satisfactory
Round 2
Reviewer 1 Report
The revised version of the manuscript still has some minor issues concerning English syntax and clarity hence it can be improved by minor revision.
Minor concerns:
The phrase in line 14: “…Chromosomal instability (CIN) is recognized as the main cause of aneuploidy but both can co-exist”... better be excluded or rephrased because its unclear.
The phrase in lines 19-21: “…While low levels of CIN can foster malignant disease, high levels frequently trigger cell death, which supports the “aneuploidy paradox” that refers to the intrinsically negative impact of an aberrant karyotype on cellular fitness…” is misleading and need to be rephrased as “…While low levels of CIN can foster malignant disease, high levels frequently trigger cell death, which supports the “aneuploidy paradox” that refers to the intrinsically negative impact of a highly aberrant karyotype on cellular fitness…”
Lines 34-36: “…Hence, cell cycle progression involves passage through multiple tightly regulated checkpoints, some of them key to prevent chromosome segregation errors and CIN…” that could be more precisely written as: “…Hence, cell cycle progression involves passage through multiple tightly regulated checkpoints, some of them key to prevent chromosome segregation errors and structural CIN…”
Line 36: “…theses…” change to: “…these…”
Figure 1 panel 3: May be significantly improved if it might include two different scenarios of mitotic slippage a) the existing figure panel depicting a multipolar mitosis driven by non-clustered centrosomes better corrected to lead to cell death by mitotic catastrophe and not to CIN and aneuploidy b) this panel may benefit if expanded with a novel graphic depicting an additional scenario of clustered centrosomes and endoreduplicated chromosomes (diplochromosomes) that leads to WGD and has the potential to undergo further divisions that may lead to additional structural or numerical CIN and aneuploidy.
Lines 98-99: “…As mentioned, mitotic arrest following lack or faulty sister-chromatid to kinetochore attachment depends on the inhibition of Cyclin B1 degradation…” better written as “…As mentioned, mitotic arrest following lack or faulty attachment of sister-chromatid to kinetochore, depends on the inhibition of Cyclin B1 degradation…”
Line 114: What PTM stands for?
Line 159: What OMM stands for?
Line 250: “…divers…” The authors mean diverse?
Line 250: “…even though numerical aneuploidies do not unequivocally lead to p53 activation (65)…” better change to “…whereas aneuploidies do not unequivocally lead to p53 activation (65)…” or “…whereas numerical CIN does not unequivocally lead to p53 activation (65)…”
Lines 271-273 “…DNA damage, as a consequence of chromosomal mis-segregation, either by DNA-double-stranded breaks or micronuclei formation and chromothripsis, are contributing events (66, 67)…” Do the authors mean “…DNA damage, as a consequence of chromosomal mis-segregation, DNA-double-stranded breaks or micronuclei formation and chromothripsis, are contributing events (66, 67)…” ?
Lines 273-274: references for micronuclei needed
Lines 289, 302 and 429: better replace “…CIN…” with “…numerical CIN…”
Line 331: change “…aneuploidy…” to “…aneuploid…”
Line 331: change “…promotes…” to “… promote…”
Line 452: better change “…It would be easy to conclude, both pathways are…” to “… It would be easy to conclude that both pathways are…”
Line 461: better change “…down…” to “…Down…”
Author Response
We want to thank this referee for taking a second look.
We have followed the suggestions and corrected minor mistakes, as pointed out under "minor concerns"
We also adopted Figure 1 and hope that this further improves clarity
We hope that these changes are now satisfactory